# Dahl Salt-Resistant Rat Is Protected against Hypertension during Diet-Induced Obesity

**DOI:** 10.3390/nu14183843

**Published:** 2022-09-16

**Authors:** Soyung Lee, Sungmin Jang, Jee Young Kim, Inkyeom Kim

**Affiliations:** 1Department of Pharmacology, School of Medicine, Kyungpook National University, Daegu 41944, Korea; 2Cardiovascular Research Institute, Kyungpook National University, Daegu 41944, Korea; 3BK21 Plus KNU Biomedical Convergence Program, Department of Biomedical Science, Kyungpook National University, Daegu 41944, Korea

**Keywords:** Dahl salt-resistant rat, Dahl salt-sensitive rat, hypertension, obesity, perivascular adipose tissue

## Abstract

A high-fat diet (HFD) frequently causes obesity-induced hypertension. Because Dahl salt-resistant rats are protected against hypertension after high-salt or high-fructose intake, it is of interest whether this model also protects against hypertension after diet-induced obesity. We tested the hypothesis that Dahl salt-resistant rat protects against hypertension during diet-induced obesity. Dahl salt-sensitive (SS) and Dahl salt-resistant (SR) rats were fed a HFD (60% fat) or a chow diet (CD; 8% fat) for 12 weeks. We measured blood pressure using the tail-cuff method. The paraffin sections of thoracic perivascular adipose tissue (tPVAT) were stained with hematoxylin/eosin and trichrome. The expression of genes in the tPVAT and kidneys were measured by reverse transcription-quantitative polymerase chain reaction. The HFD induced hypertension in SS (*p* < 0.01) but not SR rats, although it increased body weight gain (*p* < 0.05) and tPVAT weight (*p* < 0.01) in both rats. The HFD did not affect the expression of genes related to any of the adipocyte markers in both rats, although SR rats had reduced beige adipocyte marker Tmem26 levels (*p* < 0.01) and increased anti-inflammatory cytokine adiponectin (*p* < 0.05) as compared with SS rat. The HFD did not affect the mRNA expression of contractile factors in the tPVAT of SS and SR rats. SR rats are protected against hypertension during diet-induced obesity. This result implies that the genetic trait determining salt sensitivity may also determine fructose and fat sensitivity and that it is associated with the prevention of hypertension.

## 1. Introduction

In many countries, the prevalence of obesity is increasing, and one-third of the worldwide population is described as obese or overweight [1]. Obesity is currently considered a pandemic and is frequently associated with metabolic syndromes such as hypertension, insulin resistance, and abnormal blood lipid levels [2,3]. A high-fat diet (HFD) increases excessive body fat storage and is a major factor in obesity [4]. This diet represents a risk factor for the development and/or worsening of several chronic diseases, including cardiovascular diseases [5].

Perivascular adipose tissue (PVAT) is the adipose tissue that surrounds the blood vessels. PVAT differs based on the species and anatomic location and is generally classified as brown (thermoactive adipocytes), beige (brown-like adipocytes), or white adipose tissue (lipid-storing adipocytes) [6]. PVAT releases a wide range of adipokines, vasoactive and pro-/anti-inflammatory mediators that influence vascular function in a paracrine manner [7]. PVAT plays a beneficial role by releasing anticontractile factors such as hydrogen sulfide, adiponectin, and nitric oxide (NO) in equilibrium. A great number of contractile factors are produced in obesity, such as angiotensin II, chemerin, serotonin, dopamine, norepinephrine, calpastatin, and so on [8]. Moreover, obesity is frequently associated with structural and functional alterations in PVAT, leading to vascular dysfunction that involves the endothelium and smooth muscle as well as increased overall cardiovascular risk [9].

Hypertensive patients account for 25% of all patients with chronic kidney disease (CKD), and hypertensive patients with obesity are at the greatest risk for developing CKD [10,11]. The development of obesity leads to significant lipid deposits around or within non-adipose tissues and organs (ectopic fat), which impairs both tissue and organ function [12]. Obesity also affects the kidney, which has assigned roles in dyslipidemia, the production of adipokines and angiotensin II, oxidative stress, hyperfiltration, immune activation, and lipotoxicity [13,14]. The activation of the renin-angiotensin-aldosterone system (RAAS) is the main mechanism by which obesity induces the development of high blood pressure [15]. Because experimental results from animals and humans have suggested the activation of the RAAS in hypertension with obesity, the RAAS is considered important in controlling blood pressure in obesity [16].

A recent study showed that a HFD increases blood pressure in the Dahl salt-sensitive (SS) rat model [17]. The SS rat has been regarded as the most popular model of human salt-sensitive hypertension. Salt-sensitive hypertension is more likely to cause multiple organ damage and results in a higher prevalence of cardiovascular and renal diseases among hypertensive subjects [18]. Several studies have shown that the Dahl salt-resistant (SR) rat remains normotensive in resistance to diet-induced hypertension, such as with high salt or high fructose intake, as compared with the SS rat [19]. However, it is of interest whether the SR rat is protected against hypertension after diet-induced obesity. Therefore, we tested the hypothesis that SR rat is protected against hypertension during diet-induced obesity.

## 2. Materials and Methods

### 2.1. Animals

The Institutional Review Board of Kyungpook National University approved the care and use of animals (approval No. 2021-0192). Every effort was made to minimize both the number of animals used and their suffering. Seven-week-old male SS rats (DIS/EisSlc; Dahl-Iwai S) and SR rats (DIR/EisSlc; Dahl-Iwai R) were purchased from Japan SLC, Inc. (Hamamatsu, Shizoka, Japan, n = 6). The rats were acclimatized for a couple of weeks before being randomly assigned into one of two groups fed with a chow diet (CD; containing 8.6% fat; SAFE, Paris, France) [20] or a HFD (60% fat, Research Diets, Inc., New Brunswick, NJ, USA) and allowed free access to food and tap water for 12 weeks. The body weight of the rats was monitored weekly for 12 weeks using an electronic scale (KB-5000, A&D KOREA Ltd., Seoul, Korea). Caloric intake was calculated by multiplying food intake (g/day) by 3.1 kcal/g for the CD groups and 5.2 kcal/g for the HFD groups, respectively. The rats were anesthetized with sodium pentobarbital (50 mg/kg intraperitoneally) for sacrifice, after which, tissues were obtained, frozen in liquid nitrogen, and stored at −80 °C until further study.

### 2.2. Blood Pressure Measurements

We measured the systolic blood pressure (SBP) and diastolic blood pressure (DBP) of the rats using the tail-cuff method. Rats were preheated on a hotplate at 35 °C for 10 min and then placed in plastic restrainers. We attached a cuff with a pneumatic pulse sensor around the tail. The CODA system (Kent Scientific Corporation, Torrington, CT, USA) was used to record blood pressure values with heating. Blood pressure measurements were averaged from at least ten consecutive readings obtained from each rat [20].

### 2.3. Histological Analysis

After the thoracic aorta including PVAT had carefully been excised, the PVAT was isolated from the thoracic aorta and weighed. For staining hematoxylin and eosin (H&E) and trichrome, thoracic aorta and the surrounding thoracic PVAT (tPVAT) were fixed in 4% formalin overnight and then dehydrated and embedded in paraffin using conventional methods. Paraffin-embedded samples were sectioned to a thickness of 2.0 μm. After staining, slides were examined with light microscopy. We used ImageJ software (National Institutes of Health, Bethesda, MD, USA) to measure the area of the adipocytes. The calculated areas were multiplied by a conversion factor to determine the cross-sectional area of the adipocytes in μm^2^.

### 2.4. Reverse Transcription Quantitative Polymerase Chain Reaction (RT-qPCR)

Tissues (around 100 mg) were homogenized in liquid nitrogen with a glass homogenizer. Total RNA was extracted using QIAzol^®^ Lysis Reagent (QIAGEN Science, Germantown, MD, USA) according to the manufacturer’s instructions. RNA was converted into cDNA using the RevertAid™ first-strand cDNA synthesis kit (ThermoFisher Scientific, Waltham, MA, USA) according to the manufacturer’s instructions. Next, we conducted a quantitative polymerase chain reaction using an ABI Prism 7500 sequence detection system (Applied Biosystems, Foster City, CA, USA). The reaction solution (20 μL) contained 10 μL of SYBR Green master mix (New England Biolabs, Ipswich, MA, USA), 4 μL of cDNA, and 6 μL of primer set (200 nmol/L). The PCR reactions were conducted as follows: 2 min at 50 °C, 10 min at 95 °C, and 40 cycles at 95 °C for 15 s, followed by 1 min at 60 °C. The relative expression levels were determined as the Δcycle threshold (ΔCt) [19]. All primer sets used in the present study are shown in Supplemental Appendix A.

### 2.5. Measurement of Cytokine Levels

We used enzyme-linked immunosorbent assay (ELISA) kits to analyze the following in PVAT: interleukin-6 (IL-6; BMS625; ThermoFisher Scientific), tumor necrosis factor α (TNFα; ab100785; abcam, Cambridge, UK), and adiponectin (RRP300; R&D Systems, Inc., Minneapolis, MN, USA). The optical density was read at 450 nm. The concentrations of IL-6, TNFα, and adiponectin were calculated in accordance with the standard curves.

### 2.6. Statistics

Data were expressed as mean ± standard error of mean (SEM). Statistical analyses were performed using Graph Pad Prism 7 (GraphPad Software, San Diego, CA, USA) with a p value of <0.05 considered significant.

## 3. Results

### 3.1. HFD Increased Blood Pressure Only in SS Rats Regardless of Accelerated Body Weight Gain in Both Rats

To determine whether an HFD had an effect on blood pressure and body weight, we measured the SBP, DBP, body weight, water intake, food intake, and calorie intake in the rats during the HFD for 12 weeks. The HFD significantly increased the blood pressure in SS rats but not SR rats (Figure 1A). Compared with the CD groups, the HFD steadily increased body weight (Figure 1B). The HFD did not affect water intake (Figure 1C). Although the HFD groups ate less than the CD groups did (Figure 1D), there was no difference in calorie intake between the CD and HFD groups (Figure 1E).

### 3.2. HFD Increased the Adipocyte Area in the SS Rats Regardless of Increased tPVAT Weight in Both Rats

We measured the tPVAT weight and adipocyte area to evaluate the effect of the HFD on tPVAT characteristics. The HFD significantly increased fat deposits and tPVAT weight in both SS and SR rats (Figure 2A,B). H&E staining of tPVAT revealed that the HFD increased the adipocyte area in SS but not SR rats, whereas trichrome staining revealed that the HFD did not affect fibrosis (Figure 2C). Quantification of the adipocyte area supported this observation (Figure 2D).

### 3.3. HFD Did Not Affect the Expression of Genes Related to Adipocyte Markers in tPVAT of SS and SR Rats

Based on the results that the HFD increased the adipocyte area only in the SS rats, we measured the expression of genes related to adipocyte markers, *Ucp1*, *Pgc1α*, and *Pparγ*, representing brown fat; *Tmem26* representing beige fat; and *Leptin* representing white fat in tPVAT from the SS and SR rats. SS rats had a higher gene expression of *Tmem26* as compared with SR rats fed a CD (Figure 3D), with no significant change in *Ucp1*, *Pgc1α*, *Pparγ*, or *Leptin* (Figure 3).

### 3.4. HFD Did Not Affect the Expression of the Inflammatory or Contractile Factors in tPVAT of SS and SR Rats

To determine whether the HFD affected the inflammatory factors (IL-6, TNFα, and adiponectin) in tPVAT, we conducted ELISA analysis (Figure 4). The HFD did not affect the proinflammatory factors (IL-6 and TNFα) in tPVAT of SS and SR rats (Figure 4A,B). The tissue level of the anti-inflammatory factor (adiponectin) in tPVAT was higher in SR rats than in SS rats fed a CD (Figure 4C). We measured the expression of genes related to contractile factors (angiotensin II, norepinephrine, chemerin, and serotonin) to investigate whether the HFD affected the contractile factors in tPVAT (Figure 5). The HFD did not affect the gene expressions of *Ace*, *Angiotensinogen*, *Tyrosine hydroxylase*, *Rarres2*, *Cmklr1*, or *Slc6a4* in tPVAT of SS or SR rats.

### 3.5. HFD Did Not Affect the Expression of Genes Related to RAAS in the Kidney of SS and SR Rats

Based on our findings that the HFD increased blood pressure only in SS rats without affecting the expression of contractile factors in tPVAT, we investigated whether the HFD increased the mRNA expression of RAAS genes in the kidney. The HFD did not increase the mRNA expression of *Renin*, *Ace*, *Ace2*, or *Angiotensinogen* (Figure 6), nor that of angiotensin receptors (*At1ar*, *At1br*, *At2r*, *Mas1*; Figure 7). We also investigated the effect of HFD on the mRNA expression of NADPH oxidase-related factors in the kidneys of SS and SR rats. These data showed that there were no differences in gene expression between the CD and HFD groups (Appendix A).

## 4. Discussion

These results of this study demonstrate that SR rats are protected against hypertension during diet-induced obesity. We found that a HFD did not induce hypertension in SR rats, unlike in SS rats, although it increased both body weights and tPVAT weights. The HFD did not affect the mRNA expression of adipocyte markers or contractile factors in tPVAT of the SS and SR rats. There were no significant differences in the mRNA levels of RAAS genes in the kidneys of SS and SR rats before and after the HFD.

Salt sensitivity is defined as a decrease in mean arterial blood pressure of >5 mmHg during a low-sodium intervention or an increase of >5 mmHg during a high-sodium intervention, whereas salt resistance is defined as a change in mean arterial blood pressure of <5 mmHg during low-sodium or high-sodium interventions. The Genetic Epidemiology Network of Salt Sensitivity (GenSalt) study conducted in rural northern China demonstrated that ~32.4% of Chinese adults are sodium sensitive [21]. Physical activity and dietary potassium intake were associated with reduced sodium sensitivity, whereas older age, female gender, Black race, obesity, metabolic syndrome, and elevated BP were associated with increased sodium sensitivity [21]. In 65% of individuals with high blood pressure, the amount of salt intake had a greater effect on blood pressure in patients with metabolic syndromes such as hypertension, diabetes, and obesity than in those without the syndrome [22]. This model showed similar results in a previous study in that SS rats but not SR rats developed hypertension after high-fructose intake [19]. Similarly, SBP increased in spontaneous hypertensive rats (SHRs) after 12 weeks of a HFD, whereas SBP did not begin to increase in Wistar–Kyoto rats (WKY) until 12 weeks of a HFD [23]. In the present study, we characterized the development of hypertension in SS and SR rats by monitoring SBP, DBP, body weight, and food intake during a HFD (Figure 1). The HFD increased the SBP in the SS but not in the SR rats, although the HFD caused diet-induced obesity and tPVAT enlargement in both rats. Therefore, SR rats have a unique trait in that they are resistant to the development of diet-induced hypertension.

At a given renal perfusion pressure, the SR rats had higher renal blood flow with lower resistance than the SS rats did [24] because signaling via NO is more dominant than that via O2− in the thick ascending limb cells of the renal tubule [25]. However, in the SS rats, signaling via O2− is more dominant than that via NO in the thick ascending limb cells of the renal tubule, which results in higher vascular resistance with constriction of the renal blood vessels. Because the genetic traits that determine salt resistance in SR rats may also determine fructose or fat resistance, it remains to be investigated which genes of the SR rats determine salt resistance.

Obesity-induced hypertension has been linked to increased coagulability, endothelial dysfunction, and inflammation [26], as well as more conventional risk factors for cardiovascular diseases such as insulin resistance, hypertension, and atherogenic dyslipidemia consisting of hypertriglyceridemia, high low-density lipoprotein cholesterol particles, and suboptimal high-density lipoprotein cholesterol levels [27]. Hypertension mediates abnormal kidney function and vascular pathology. The mediators are physical compression of the kidneys by fat in and around the kidneys, activation of the RAAS, and increased activity of the sympathetic nervous system [28]. Obesity is associated with volume expansion of extracellular fluid, which increases blood flow in many tissues and, in turn, increases venous return and cardiac output [29]. Therefore, obesity is associated with functional vasodilation that is probably due to the increased metabolic rate and higher tissue oxygen consumption [28].

Because visceral fat and PVAT increase the risk for cardiovascular diseases [30], an alteration in body fat distribution might have a significant impact on high blood pressure. In states of obesity, as the PVAT mass increases, the contractile and the proinflammatory cytokine TNFα increase, whereas anticontractile factors significantly decrease [31]. We observed no significant differences between the SS and SR rats fed a CD or HFD in the mRNA expression of angiotensin II-related genes such as *Ace* and *angiotensinogen*, a norepinephrine-related gene such as *tyrosine hydroxylase*, chemerin-related genes such as *Rarres2* and *Cmklr1*, or a serotonin-related gene such as *Slc6a4* in tPVAT (Figure 5). Although there were no significant differences in the levels of the proinflammatory cytokines IL-6 and TNFα, the anti-inflammatory cytokine adiponectin was higher in the SR rats than in the SS rats fed a CD (Figure 4). PVAT-derived factors such as IL-6 [32] and TNFα [33] had a contractile or an anticontractile effect depending on the microenvironment. Regardless of whether the rats were fed a CD or HFD, there were no significant differences in the mRNA expression or cytokine levels in the SS and SR rats. Therefore, the factors with anticontractile or contractile effects may exert a contractile effect in obese SS rats.

An HFD tends to increase urinary protein excretion with marginal kidney damage and further accelerates salt-induced proteinuria and renal histological aggravation [34]. Although several studies have reported that HFD-induced hypertension increases the expression of RAAS in the kidneys of rodent models [35,36], no differences were found in the expression of the RAAS components (*Renin, Ace, Ace2, Angiotensinogen, At1ar, At1br, At2r,* and *Mas1*) in kidneys of SS and SR rats fed a CD or HFD. The increase in SBP did not correlate with the mRNA expression of the RAAS components in the kidney; these findings suggest that the mRNA expression of the RAAS components did not affect the BP in the SS and SR rats. Another study demonstrated that a HFD for 12 weeks did not induce a definite diabetic state despite hypertension in SHRs and suggested that a HFD for 24 weeks induced definite dyslipidemia and insulin resistance as well as systemic RAAS activation [12].

In particular, Nox2 represents endothelial and vascular dysfunction in metabolic disease and hypertension [37]. HFD-induced hypertension shows that endothelial Nox2-derived superoxide plays a critical role in endothelial dysfunction, whereas Nox2 deficiency or pharmacologic Nox2 inhibition protects against vascular oxidative stress. Nox1 and Nox2 primarily produce O_2_^∙−^. Nox4 is a functional source of reactive oxygen species generation in the mitochondria of the kidney cortex, wherein mitochondrial superoxide dismutase (SOD) effectively dismutates Nox4-derived superoxide to H_2_O_2_ [38]. In our study, *Nox2* was markedly higher in the kidney of SR rats than in SS rats, with no significant changes in *Nox1, Nox3,* or *Nox4* (Appendix A).

The constriction of the mesenteric vascular bed provides an important contribution to the total peripheral resistance [39]. Structural or functional alterations in the mesenteric vascular bed contribute to the hypertensive process [40]. The mesenteric arteries of SHR express a greater density of calcium channels than those of WKY, which exhibited exaggerated constrictor responses to a variety of stimuli in SHR [41,42]. A high-salt diet was shown to decrease SOD activity in the mesenteric arteries of SS rats [43]. An HFD induced endothelial dysfunction in the mesenteric vascular bed due to a proinflammatory and contractile state of the PVAT [44].

This study has some limitations. First, we measured SBP and DBP by the tail-cuff method. Second, PVATs at different anatomical locations present different phenotypes [1]. The brown adipose tissues in the thoracic aorta were not whitened by diet-induced obesity/inflammation and did not have the characteristics of white adipocytes [45,46]. Although the HFD increased the body weights and tPVAT weights as well as adipocyte areas of the rats, it did not affect the properties (Figure 2 and Figure 3). Although SS rats had similar cellularity in white and brown adipocytes as compared to SR rats, SS rats had higher cellularity in beige adipocytes than SR rats, which were not affected by the HFD. It would be of interest to determine whether an HFD affects abdominal or mesenteric PVAT, as the mesenteric vascular bed plays a key role in blood pressure regulation. Finally, although the HFD groups and CD groups had a similar calorie intake, it is possible that reduced energy expenditure contributed to increases in body weight in the HFD groups [17].

In summary, the blood pressure of the SR rats fed an HFD was not affected by the fat accumulation, properties of tPVAT, or RAAS in the kidney. Therefore, our results indicate that the SR rat is protected against hypertension during diet-induced obesity. This finding implies that the genetic trait that determines salt sensitivity may also determine fructose and fat sensitivity and that it is associated with the prevention of hypertension.

## Figures and Tables

**Figure 1 nutrients-14-03843-f001:**
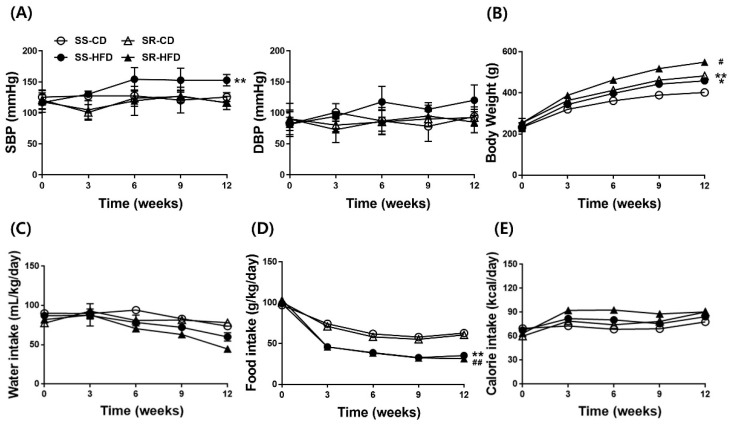
The effects of a high-fat diet on systolic blood pressure (SBP), diastolic blood pressure (DBP), body weight, water intake, food intake, and calorie intake in Dahl salt-sensitive (SS) and salt-resistant (SR) rats. SS and SR rats were fed either a chow diet (CD, 8.6% fat) or a high-fat diet (HFD, 60% fat) for 12 weeks. (**A**) The HFD increased the SBP in SS rats but not in SR rats. (**B**) The HFD increased body weight gain in both the SS and SR rats. (**C**–**E**) Water intake, food intake, and calorie intake are shown. As food intake decreased in the HFD groups, there was no difference in calorie intake between the CD and HFD groups. Graph, mean ± SEM of 6 independent experiments. Two-way analysis of variance followed by Tukey’s *post hoc* multiple comparisons test. * *p* < 0.05 and ** *p* < 0.01 vs. the SS CD group. # *p* < 0.05 and ## *p* < 0.01 vs. the SR CD group. ○, SS-CD; ●, SS-HFD; △, SR-CD; ▲, SR-HFD.

**Figure 2 nutrients-14-03843-f002:**
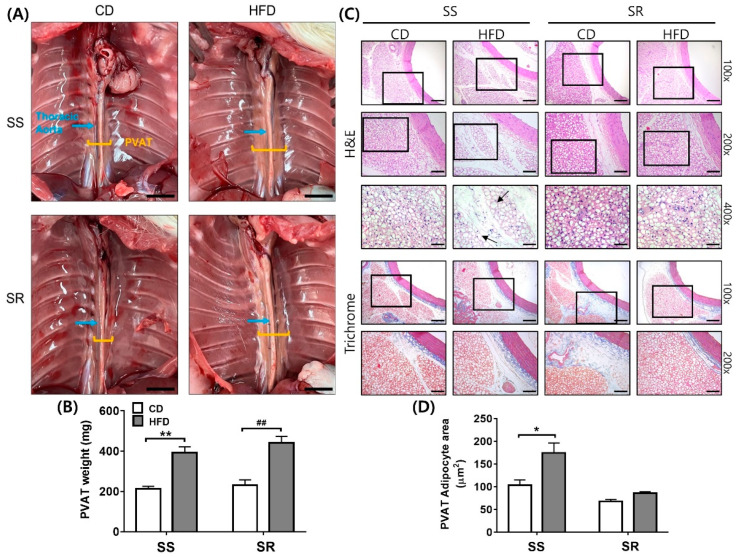
The effects of a high-fat diet on fat deposits, weight, histological characteristics, and adipocyte area of the thoracic perivascular adipose tissues (tPVAT) in SS and SR rats. (**A**) Representative pictures visualizing tPVAT from the SS and SR rats fed a CD or HFD for 12 weeks. The HFD increased fat deposits (**A**) and PVAT weight (**B**) in both the SS and SR rats. (**C**,**D**) Representative microscopic images of the thoracic aortas and tPVAT from the SS and SR rats fed a CD or HFD for 12 weeks. The thoracic aortas and tPVAT sections were stained with hematoxylin and eosin (H&E, upper) and trichrome (lower) stains. The scale bars for 100×, 200×, and 400× magnification were 200, 100, and 50 μm, respectively. The HFD increased the PVAT adipocyte area (black arrow) in SS rats but not SR rats. Graph, mean ± SEM of 6 independent experiments. Two-way analysis of variance followed by Tukey’s post hoc multiple comparisons test. * *p* < 0.05 and ** *p* < 0.01 vs. the SS CD group. ## *p* < 0.01 vs. the SR CD group.

**Figure 3 nutrients-14-03843-f003:**
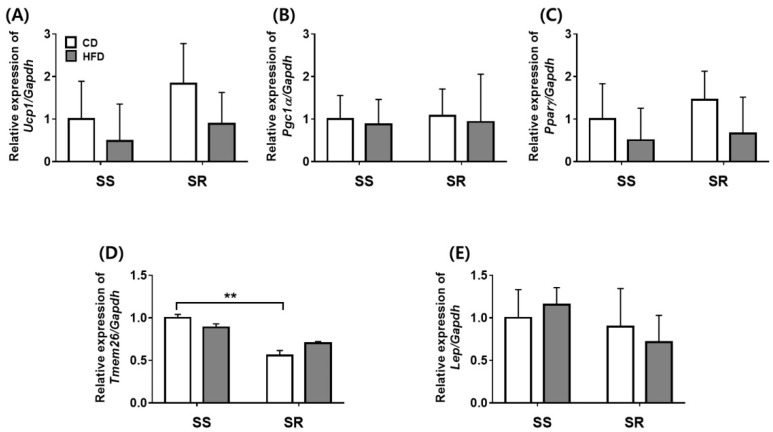
The effects of a high-fat diet on the expression of adipocyte marker genes in PVAT from SS and SR rats. The mRNA expression of brown adipocyte-related genes such as *uncoupling protein 1* (*Ucp1*, **A**), *peroxisome proliferator-activated receptor γ coactivator 1-α* (*Pgc-1α*, **B**), and *peroxisome proliferator-activated receptor γ* (*Pparγ*, **C**), a beige adipocyte-related gene such as *transmembrane protein 26* (*Tmem26*, **D**), and a white adipocyte-related gene such as *Leptin* (**E**) were measured by RT-qPCR in PVAT from SS and SR rats fed a CD or HFD for 12 weeks. Graph, mean ± SEM of 6 independent experiments. Two-way analysis of variance followed by Tukey’s post hoc multiple comparisons test. ** *p* < 0.01 vs. the SS CD group.

**Figure 4 nutrients-14-03843-f004:**
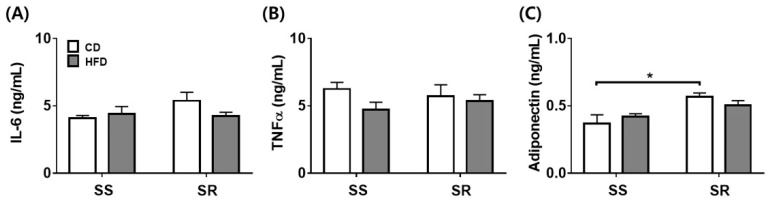
The effects of a high-fat diet on adipose tissue derived inflammatory factors in PVAT from SS and SR rats. Tissue levels of proinflammatory cytokines (interleukin-6 [IL-6, **A**] and tumor necrosis factor α [TNFα, **B**]) and an anti-inflammatory cytokine (adiponectin, **C**) were detected via enzyme-linked immunosorbent assay (ELISA) in SS and SR rats fed a CD or HFD for 12 weeks. Graph, mean ± SEM of 6 independent experiments. Two-way analysis of variance followed by Tukey’s post hoc multiple comparisons test. * *p* < 0.05 vs. the SS CD group.

**Figure 5 nutrients-14-03843-f005:**
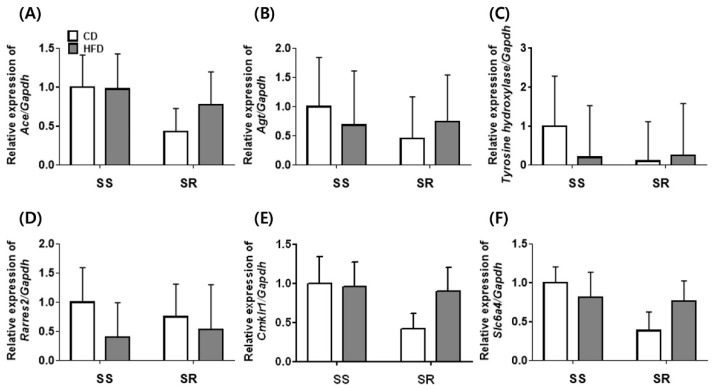
The effects of a high-fat diet on the expression of genes related to contractile factors in PVAT from SS and SR rats. The mRNA expression of angiotensin II-related genes such as *angiotensin-converting enzyme* (*Ace*, **A**) and *angiotensinogen* (*Agt*, **B**), a norepinephrine-synthesizing gene such as *tyrosine hydroxylase* (**C**), chemerin-related genes such as *rarres2* (**D**) and chemerin receptor 23 (*Cmklr1*, **E**), and a serotonin-related gene such as serotonin transporter (*Slc6a4*, **F**) were measured by RT-qPCR in PVAT from SS and SR rats fed a CD or HFD for 12 weeks. Graph, mean ± SEM of 6 independent experiments. Two-way analysis of variance followed by Tukey’s post hoc multiple comparisons test.

**Figure 6 nutrients-14-03843-f006:**
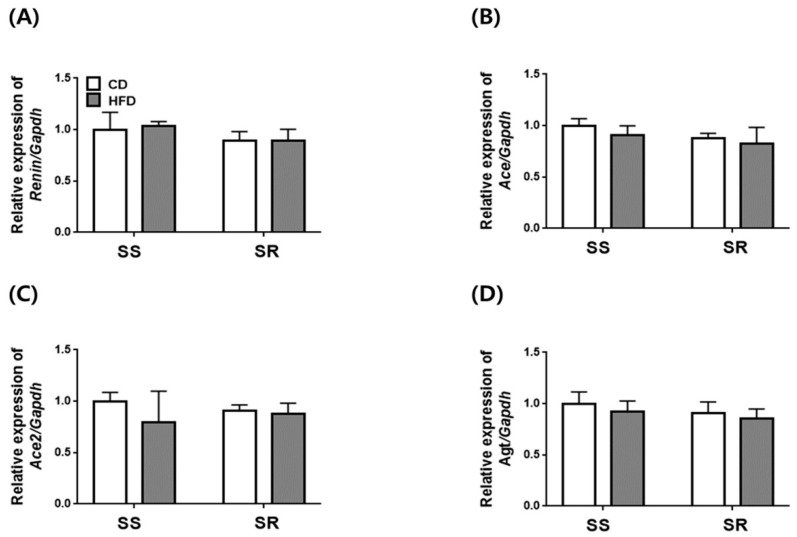
The effects of a high-fat diet on the expression of genes related to angiotensin II in kidneys from SS and SR rats. The mRNA expressions of angiotensin II-related genes such as *Renin* (**A**), *Ace* (**B**), *Ace2* (**C**), *Agt* (**D**) were measured by RT-qPCR in the kidneys of SS and SR rats fed a CD or HFD for 12 weeks. Graph, mean ± SEM of 6 independent experiments. Two-way analysis of variance followed by Tukey’s post hoc multiple comparisons test.

**Figure 7 nutrients-14-03843-f007:**
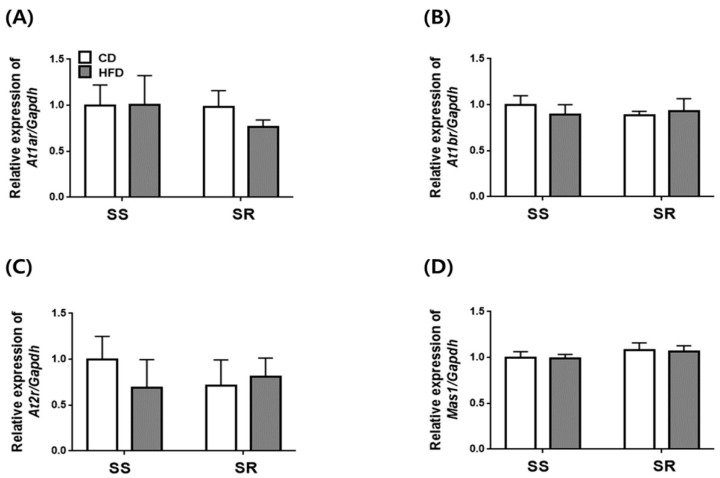
The effects of a high-fat diet on the expression of genes related to angiotensin II receptor in kidney from SS and SR rats. The mRNA expression of angiotensin receptor related genes such as *angiotensin II type 1a receptor* (*At1ar*, **A**), *angiotensin II type 1b receptor* (*At1br*, **B**), *angiotensin II type 2 receptor* (*At2r*, **C**), and *mas1 proto-oncogene* (*Mas1*, **D**) were measured by RT-qPCR in kidney from SS and SR rats fed a CD or HFD for 12 weeks. Graph, mean ± SEM of 6 independent experiments. Two-way analysis of variance followed by Tukey’s post hoc multiple comparisons test.

## Data Availability

Informed consent was obtained from all subjects involved in the study.

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
