# Peer review of "Dahl Salt-Resistant Rat Is Protected against Hypertension during Diet-Induced Obesity"

_nutrients, 2022, doi:10.3390/nu14183843_

Round 1

Reviewer 1 Report

Dear all,

The manuscript fits with the aim of the journal, and the subject reveals good content for researchers and professionals in the nutrition field. However, some minor points are listed below:

How many rats were involved in the study?

In line 79, regarding the high-fat diet (HFD), (60% fat, Research Diets, Inc., New Bruns-79 wick, NJ, USA). Could you clarify other diet components! And can you show how much kcal percentage from fat (e.g. XX% kcal from fat).

You mentioned body weight In lines 22,126, 128, 129, 132, 136, 139, 234, 254, 326, 333. At the same time, you didn’t mention in the Materials and Methods how did you monitor body weight, please add this to 2.1. Animals section.

How were rats housed? is food and water were free or locked-up access?

In line 90, Blood pressure measurements were averaged from at least ten consecutive readings obtained from each rat.  Ten readings from each rat are an excellent work. But why you measured only systolic blood pressure (SBP) and didn’t measure diastolic blood pressure (DBP)?

In lines 243 to 246, this section needs reference.

What clinical significance does this work have? Must be expanded upon and clarified.

line 322, at limitations of the study, you spoke about and stated, if possible, include a section that will result in future scientific research.

Please emphasize the clinical relevance of your work in the conclusion. How does this work relate to the ongoing clinical work?

Best regards,

Reviewer 2 Report

Journal: Nutrients

Manuscript title:  Dahl salt-resistant rat is protected against hypertension during 2

diet-induced obesity

This study evaluates the effect of SR or SS-ingestion in HFD-induced hypertension in rats. In my opinion a good study, but needed some modifications:

1.      This manuscript with a plagiarism rate of 43% without references and citations which is too

Abstract:

2.       The abstract must contain all the results with the statistic

Methods

3.      The study design where these administrations were borrowed need to be cited to give credit to previous work where the author based his methods, this should be done throughout the methodology unless you are not the first to use this fermented milk in such doses

4.      Doses used to justify or cite references

Results

5.      histograms are too small and unclear

6.      Same remark for the histological study, it is necessary to note the anomalies on the photos

Round 2

Reviewer 2 Report

In my opinion this version of manuscript is perfectly revised and can accepted for publication